# Impact of Alcohol and Smoking on Outcomes of HPV-Related Oropharyngeal Cancer

**DOI:** 10.3390/jcm11216510

**Published:** 2022-11-02

**Authors:** Yu-Hsuan Lai, Chien-Chou Su, Shang-Yin Wu, Wei-Ting Hsueh, Yuan-Hua Wu, Helen H. W. Chen, Jenn-Ren Hsiao, Ching-Hsun Liu, Yi-Shan Tsai

**Affiliations:** 1Department of Oncology, National Cheng Kung University Hospital, College of Medicine, National Cheng Kung University, Tainan 704302, Taiwan; 2Clinical Innovation and Research Center, National Cheng Kung University Hospital, College of Medicine, National Cheng Kung University, Tainan 704302, Taiwan; 3Institute of Clinical Medicine, College of Medicine, National Cheng Kung University, Tainan 701401, Taiwan; 4Department of Otolaryngology, National Cheng Kung University Hospital, College of Medicine, National Cheng Kung University, Tainan 704302, Taiwan; 5Department of Pathology, National Cheng Kung University Hospital, College of Medicine, National Cheng Kung University, Tainan 704302, Taiwan; 6Department of Medical Imaging, National Cheng Kung University Hospital, College of Medicine, National Cheng Kung University, Tainan 704302, Taiwan

**Keywords:** alcohol, smoking, betel nut, human papillomavirus (HPV), oropharyngeal cancer, treatment-effect modifier, prognostic factor

## Abstract

Background: The aim of this study was to evaluate the impact of adverse lifestyle factors on outcomes in patients with human papillomavirus (HPV)-related oropharyngeal squamous cell carcinoma (OPSCC). Methods: From 2010 to 2019, 150 consecutive non-metastatic OPSCC patients receiving curative treatment in our institution were retrospectively enrolled. HPV positivity was defined as p16 expression ≥75%. The effects of adverse lifestyle factors on overall survival (OS) and disease-free survival (DFS) on OPSCC patients were determined. Results: The median follow-up duration was 3.6 years. Of the 150 OPSCCs, 51 (34%) patients were HPV-positive and 99 (66%) were HPV-negative. The adverse lifestyle exposure rates were 74.7% (*n* = 112) alcohol use, 57.3% (*n* = 86) betel grid chewing, and 78% (*n* = 117) cigarette smoking. Alcohol use strongly interacted with HPV positivity (HR, 6.00; 95% CI, 1.03–35.01), leading to an average 26.1% increased risk of disease relapse in patients with HPV-positive OPSCC. Heavy smoking age ≥30 pack-years was associated with increased risk of death (HR, 2.05; 95% CI, 1.05–4.00) and disease relapse (HR, 1.99; 95% CI, 1.06–3.75) in OPSCC patients. In stratified analyses, the 3-year absolute risk of disease relapse in HPV-positive OPSCC patients reached up to 50% when alcohol use and heavy smoking for ≥30 pack-years were combined. Conclusions: Alcohol acted as a significant treatment-effect modifier for DFS in HPV-positive OPSCC patients, diluting the favorable prognostic effect of HPV positivity. Heavy smoking age ≥30 pack-years was an independent adverse prognostic factor of OS and DFS in OPSCC patients. De-intensification treatment for HPV-related OPSCC may be avoided when these adverse lifestyle factors are present.

## 1. Introduction

Over the past few decades, the prevalence of human papillomavirus (HPV)-related oropharyngeal squamous cell carcinoma (OPSCC) has increased rapidly—particularly in high-income countries [1,2]. Unlike other head and neck squamous cell carcinomas (HNSCCs), HPV-related OPSCCs have distinct clinical presentations: the patients tend to be younger and the cancers are less associated with smoking and more associated with primary tonsillar tumors and cystic cervical lymph node metastasis [3]. HPV-16 accounts for at least 85% of all HPV-related OPSCCs [4]. Two HPV oncogenes, E6 and E7, are key drivers of HPV-mediated carcinogenesis. E6 and E7 involve increased degradation of the tumor suppressor proteins p53 and Rb, respectively, resulting in the loss of cell-cycle checkpoint activation in response to DNA damage and uncontrolled licensing of DNA replication—which together result in genomic instability and resistance to apoptosis [5,6]. p16 is upregulated during the process of E7-directed epigenetic reprogramming [7]. Thus, p16 overexpression is a surrogate marker for HPV-related OPSCC [8]. The cutoff of p16 positivity by immunohistochemistry (IHC) staining is nuclear expression ≥+2/+3 intensity and ≥75% distribution [9].

The prognostic significance of HPV status in OPSCC has been established; patients with HPV-related OPSCC have a more favorable treatment response and longer survival time than HPV-unrelated OPSCC [10,11,12]. The American Joint Committee on Cancer (AJCC) has defined HPV-positive and HPV-negative OPSCCs as separate entities because of their distinct tumor characteristics, biological behaviors, and treatment outcomes [9,13].

Asian OPSCC patients have poorer treatment outcomes than other ethnicities [14,15]. One suspected reason for this is the lower rate of HPV positivity in OPSCC, which is about 30% to 50% in Asians but 70% to 85% in Western populations [2]. Higher rates of alcohol use, betel grid chewing, and cigarette smoking (ABC lifestyle factors) in Asia also might contribute to poorer prognosis [14]. The ABC lifestyle factors are especially common in Southeast Asia—especially in low socioeconomic and less-educated populations [16]. ABC lifestyle factors usually coexist, which may contribute to a dramatically increased risk of developing HNSCC in multi-user persons compared with that in persons who have never been exposed to ABC factors [17]. Although the role of ABC lifestyle factors has been well established in the development of HNSCC [18,19], less is known about their prognostic significance in patients with HPV-positive OPSCC. Epidemiologic studies of HPV-positive OPSCC have been conducted mostly in Western countries and are therefore not generalizable to non-Western countries [20], where factors such as cultural and behavioral differences might result in different etiologies in HPV-positive OPSCC. Wider geographically based investigations are necessary to guide region-specific clinical treatments and public health policies.

As de-intensification treatment protocols in patients with HPV-positive OPSCC are currently applied [21,22,23], it is important to identify patients where such attempts may not be safe. We hypothesized that ABC lifestyle factors moderate the effects of p16 status on survival in OPSCC patients. This study aimed to evaluate the impact of ABC lifestyle factors on treatment outcomes in patients with HPV-positive OPSCC and to optimize the selection of a subgroup of HPV-positive OPSCC patients for de-intensification treatment.

## 2. Materials and Methods

### 2.1. Patients

One hundred and fifty OPSCC patients who had completed a course of curative treatment, consisting of surgery and radiotherapy (RT)-based therapy from January 2010 to October 2019, were consecutively collected and analyzed. All patients had received a complete staging work-up before treatment and were followed to determine their treatment response and survival. The exclusion criteria were: (1) other underlying malignancy or distant metastasis at the time that OPSCC was diagnosed; (2) lack of available pretreatment primary tumor specimens to re-evaluate p16 expression by IHC staining; (3) lack of pretreatment contrast-enhanced computed tomography (CT) images of the head and neck to re-evaluate clinical staging. The Institutional Review Board approved this retrospective study.

### 2.2. Demographic and Clinical Data

Patient data—including age, gender, tumor subsites, history of ABC lifestyle (alcohol consumption, betel grid chewing, cigarette smoking), smoking age (number of cumulative pack-years of smoking), treatment-related profiles (surgery, radiotherapy, chemotherapy), and outcome data—were gathered by retrospective chart review. The clinical and pathological staging that had been determined previously were re-evaluated and revised based on the seventh and eighth editions of the AJCC staging system [9,13].

HPV status was determined by re-examination of p16 nuclear expression in the pretreatment primary tumor by IHC staining. After tissue specimens from our human biobank were collected, all the slides were re-evaluated by a head and neck pathologist with 30-years’ experience to determine the HPV status. HPV positivity was defined as the presence of p16 expression in ≥75% of carcinoma cells, with nuclear reactivity on IHC staining [9].

### 2.3. Treatments

The standard primary treatments for OPSCC were surgery and RT-based therapy. Definitive concurrent chemoradiotherapy (CCRT) with a platinum-based chemotherapeutic regimen was most often used in locally advanced OPSCC. The curative-intent radiation dose to the primary tumor and grossly involved lymph nodes was 60–74 Gy in 1.8–2.2 Gy per fraction, delivered daily with intensity-modulated radiotherapy or volumetric-modulated arc therapy techniques. Induction chemotherapy was allowed before primary treatments. Adjuvant treatments after primary surgery were indicated when patients with adverse pathological features—including positive/close surgical margin, extranodal extension, pT3–pT4 disease, positive lymph node metastasis, perineural invasion, lymphovascular space invasion, or any other concern—when determined to be appropriate by multidisciplinary discussion.

### 2.4. Statistical Analysis

Baseline characteristics were presented as mean (standard deviation) for continuous variables and number (frequency) for categorical variables. The Kaplan–Meier survival method was used to depict the curves for the distribution of time to death or relapse and log-rank tests were carried out to evaluate differences between HPV-positive and HPV-negative OPSCC patients. Overall survival (OS) was defined as the date of initial treatment to the date of death or last follow-up. Disease-free survival (DFS) was defined as the date of initial treatment to the date of disease relapse (locoregional recurrence and/or distant metastasis) or death. The *p*-value of continuous variables was calculated by the two-sample t-test whereas the *p*-value of categorical variables was calculated by the chi-square test and Fisher’s exact test.

Among patients with OPSCC, univariable Cox proportional hazard models were applied to identify significant patient characteristics associated with OS and DFS—including p16 status, gender, age, clinical stage, tumor subsites, initial treatment, and ABC lifestyle factors. We hypothesized that ABC lifestyle factors could modify the effects of p16 status for OS and DFS. We used multivariable Cox proportional hazard models with patient characteristics and p16 status and lifestyle factors as interaction terms by using the stepwise variable selection method to select relevant variables for OS and DFS. The criteria for the model fitting were based on the Akaike information criterion. Furthermore, multivariate models were constructed with interaction terms that were selected by the stepwise method and significant and clinically relevant variables from univariate analyses. The multicollinearity and proportional hazard assumption of the models were checked; none of the models showed high multicollinearity and the proportional hazard assumption was met.

The stratified analyses were made according to alcohol use and smoking age to estimate 3-year and 5-year cumulative risks of disease relapse in HPV-positive patients using multivariate models. The interaction plot was depicted by HPV status and alcohol use to show changes in the cumulative risk of disease relapse, which was estimated using multivariate models, in different situations. A two-tailed *p* value < 0.05 was considered statistically significant. All statistical results were carried out with R (version 4.1.0) software and Quanta for Medical Care AI: AI Medical Platform (QOCA AIM) 2.0 version (Quanta Computer Inc., Taoyuan, Taiwan).

## 3. Results

### 3.1. Patient Characteristics

One hundred and fifty patients were analyzed in this study; 99 (66%) had HPV-negative OPSCC and 51 (34%) were HPV-positive OPSCC patients. The mean age at diagnosis of OPSCC was 54.4 years; most were locally advanced OPSCCs. The ABC lifestyle exposure rates were: 112/150 (74.7%) patients showed alcohol consumption, 86/150 (57.3%) betel grid chewing, and 117/150 (78%) cigarette smoking. More than half of all patients (79/150; 52.7%) had concomitant ABC lifestyle exposure. Among the 150 patients, 39 (26%) were treated with primary surgery and 111 (74%) were treated with primary RT-based therapy (106 CCRT and 5 RT only). In the primary surgery group, 10 patients underwent induction chemotherapy before surgery and 35 underwent adjuvant RT/CCRT after surgery. In the primary RT-based therapy group, 22 patients underwent induction chemotherapy before definitive RT/CCRT. The baseline characteristics of the study population are summarized in Table 1.

Between the two groups of HPV-positive and HPV-negative OPSCC patients, there was no significant difference in the age distribution and clinical stage. In patients with HPV-positive OPSCC, the dominant tumor subsite was the tonsil and the majority received primary RT-based therapy. Patients with HPV-negative OPSCC had a significantly higher proportion of male gender, a higher exposure rate to ABC lifestyle factors, and a higher smoking age (Table 1).

### 3.2. Treatment Outcomes

The median follow-up time was 3.6 years. The recurrence rate was 47.3%, with 71 of the 150 patients developing disease relapse (locoregional recurrence and/or distant metastasis). The mortality rate was 46.7%, which meant that 70 of the 150 patients had expired by the time of analysis. Patients with HPV-positive OPSCC had significantly lower disease relapse and mortality rates than those with HPV-negative OPSCC (*p* < 0.001; Table 1). The 3-year overall survival (OS) and disease-free survival (DFS) rates for HPV-positive versus HPV-negative OPSCC patients were 90% versus 52% and 74.5% versus 42.9%, respectively (both *p* values < 0.0001; Figure 1).

### 3.3. Factors Affecting Overall Survival (OS)

In the multivariate analysis, the HPV status (positive: hazard ratio, 0.09; 95% CI, 0.02–0.44), clinical stage (stage IVA: hazard ratio, 2.72; 95% CI, 1.05–7.00; stage IVB: hazard ratio, 15.62; 95% CI, 5.29–46.13), and smoking age (≥30 pack-years: hazard ratio, 2.05; 95% CI, 1.05–4.00) were significant prognostic factors for OS in OPSCC patients (Table 2). There was no significant interaction between HPV positivity and ABC lifestyle factors; that is, ABC lifestyle factors were not significant treatment-effect modifiers for OS in HPV-positive OPSCC (Table 2).

### 3.4. Factors Affecting Disease-Free Survival (DFS)

In the multivariate analysis, the HPV status (positive: hazard ratio, 0.10; 95% CI, 0.02–0.49), clinical stage (stage IVA: hazard ratio, 2.87; 95% CI, 1.11–7.41; stage IVB: hazard ratio, 8.43; 95% CI, 2.83–25.08), tumor subsite (tonsil: hazard ratio, 0.46; 95% CI, 0.27–0.80), and smoking age (≥30 pack-years: hazard ratio, 1.99; 95% CI, 1.06–3.75) were significant prognostic factors for DFS in OPSCC patients (Table 3). Moreover, there was a strong interaction between HPV positivity and alcohol use (alcohol use: hazard ratio, 6.00; 95% CI, 1.03–35.01), which meant that alcohol was a significant treatment-effect modifier for DFS in HPV-positive OPSCC patients (Table 3). The presence of alcohol exposure diluted the favorable prognostic effect of HPV positivity in OPSCC patients. In a median follow-up duration of 3.6 years, alcohol use contributed to an average 26.1% increased risk of disease relapse in patients with HPV-positive OPSCC, whereas there was no risk increment in those with HPV-negative OPSCC (Figure 2). By stratification of smoking age among HPV-positive OPSCC patients with alcohol use, the 3-year absolute risk of disease relapse was 33% in those with smoking age <20 pack-years and up to 50% in those ≥30 pack-years (Table 4).

## 4. Discussion

In this study, we demonstrated that alcohol was a significant treatment-effect modifier for DFS in HPV-positive OPSCC patients. The presence of alcohol use diluted the favorable prognostic effect of HPV positivity in OPSCC patients, leading to an average 26.1% increased risk of disease relapse. A heavy smoking age of ≥30 pack-years was a poor prognostic factor of all-cause and disease-specific mortality among OPSCC patients, regardless of HPV status. On the other hand, betel grid chewing made no contribution to effect modification or the prediction of treatment outcomes in patients with OPSCC.

Confusion between treatment-effect modifiers and prognostic factors is common. Effect modifiers, also called effect moderators, are factors that influence how well an intervention affects the outcome. Prognostic factors are factors that predict the outcome of a disease [24]. Scientifically, effect modifiers must be differentiated from prognostic factors, but it is more challenging to claim that a factor is an effect modifier rather than a prognostic factor. Prognostic factors are familiar to oncologists and are used to provide patients with a more accurate prognosis, but they do not help identify which patients will respond best to a specific intervention. Effect modification has recently become of particular interest in oncology in the era of targeted therapy and immunotherapy, where the effectiveness of a treatment might largely depend on host or tumor factors [25,26]—as does HPV-related OPSCC. Due to the heterogeneous tumor behavior present in HPV-positive and HPV-negative OPSCCs, the variety of variables affecting treatment outcomes may play different roles as effect modifiers or prognostic factors. Determining the treatment-effect modifiers in HPV-positive OPSCC helps to identify subgroups of patients who respond better or worse to de-escalation treatments.

Our data indicate that alcohol is a significant treatment-effect modifier for DFS in HPV-positive OPSCC patients. Exposure to alcohol is well-known as a dominant etiologic factor of HNSCC. A large case-control study conducted by Lee et al. involved 740 HNSCC patients in Taiwan [25]; although the patients enrolled in this study were heterogeneous, the results showed a significant positive dose–response relationship between pre-diagnosis alcohol use and worse OS in HNSCC. This association was more significant for non-oral cavity HNSCC than for oral HNSCC. A possible mechanism for this is the polymorphism of the ethanol-metabolizing genes ADH1B and ALDH2, which modify the relationship between pre-diagnosis alcohol use and the OS of HNSCC patients—providing a possible biological explanation [27]. However, unlike our study, this analysis did not adjust for HPV status (due to a lack of access to the tumor tissues to test for HPV)—thus prohibiting its generalization to HPV-positive OPSCC. A recent study in Belgium demonstrated that alcohol use was a poor prognostic factor for OS in OPSCC patients and established a simplified scoring system composed of p16 status, smoking, and alcohol [28]. Another study showed that alcohol consumption was an independent factor for survival among patients with HPV-negative OPSCC rather than for those with HPV-positive OPSCC [29]. So far, relevant studies on the impact of alcohol use on HPV-positive OPSCC are scanty and contradictory. To the best of our knowledge, our study was the first to highlight the role of alcohol use as a treatment-effect modifier for HPV-positive OPSCC, which had not been previously evaluated and reported. The findings of our provided supporting evidence that p16 expression is not the only key factor for survival in OPSCC patients and demonstrated that the favorable prognostic effect of HPV positivity in OPSCC patients can be diluted by alcohol use.

Our study also demonstrated that smoking was not a treatment-effect modifier for HPV-positive OPSCC, but heavy smoking age ≥30 pack-years was a significant prognostic indicator of worse OS and DFS in OPSCC patients. A considerable amount of literature has explored the association between smoking and HPV-positive OPSCC. First, the association between smoking and the pathogenesis of HPV-positive OPSCC was suggested. The potential pathways of smoking-related carcinogenesis were likely attributed to cellular alterations and DNA damage, promoting infection by and the persistence of HPV [30]. By pooling two large head and neck cancer studies with HPV serology data, Anantharaman et al. demonstrated that smoking was consistently associated with increased risks of both HPV-positive and HPV-negative OPSCC [31]. Second, the association between smoking and treatment-related outcomes in HPV-positive OPSCC has been widely explored. The results remained somewhat controversial, with some studies reporting smoking as a poor prognostic factor independent of HPV status, which is consistent with our findings [32,33], and others reporting smoking exposure as a poor prognostic factor within the context of HPV-positive OPSCC [34,35]. Though there are some conflicting results, most studies agree that smoking is associated with worse OS and a trend towards worse DFS in HPV-positive OPSCC [36]. Third, the association between the amount of smoking and worse survival outcomes in HPV-positive OPSCC has been investigated. There is no consensus on the cutoff for high-risk smokers. Several previous studies have reported smoking age >10 pack-years as a cutoff delineating higher risk HPV-positive OPSCC patients [10,37]. Other smoking metrics reported in the literature have included smoking age >20 pack-years, >20 cigarettes daily, total pack-years, current smoking, and ever smoking—with variable prognostic effects on survival outcomes [38,39,40]. The divergence in outcomes determined by these smoking metrics might be due to heterogeneous patient populations, various sample sizes, and different lifestyle factors, with exposure influenced by different cultures. However, most studies have agreed that the heavier the smoking, the worse the survival outcomes [36]. Our study recommended a cutoff of ≥30 pack-years smoking age for risk stratification in Asian OPSCC populations. As more than three-quarters of our study population had a smoking history—with the majority being heavy smokers—and more than half also had ABC lifestyle exposure, our study populations were more reflective of current conditions among OPSCC patients in Southeast Asia [14,16,41].

Compared to previous studies, the strength of our study was the clear definition of HPV positivity as the presence of p16 expression in ≥75% of carcinoma cells, showing nuclear reactivity on IHC staining [9]. We repeated p16 immunostaining tests in all the pretreatment primary tumor tissues, which were reviewed by a 30-year-experienced head and neck pathologist to accurately discriminate between HPV positivity and negativity. Furthermore, all the pretreatment CT images of the head and neck region were reviewed by a 15-year-experienced radiologist to revise clinical staging based on the seventh and eighth editions of the AJCC staging system to accurately display disease status. Detailed ABC lifestyle exposure histories and an adequate follow-up duration (median 3.6 years) made our results more convincing. Despite the retrospective study design, the consecutive enrollment of qualified OPSCC participants made the internal validity of patient selection solid and reliable. Finally, and most importantly, this was the first study providing the new concept, with convincing evidence, that alcohol is a treatment-effect modifier for HPV positivity.

The limitation of this study was the lack of quantification of alcohol consumption. Detailed quantification of alcohol consumption can include information on drinking status, frequency, the level of drinking, and drink-years [27]. Due to the retrospective nature of this study, our information on alcohol use relied on medical record reviews. There might be some inaccurate reporting due to recall bias when taking histories, or potential falsification of alcohol history due to guilt or shame. Further studies would benefit from including more objective measures of alcohol quantification such as questionnaires or prospective study designs.

## 5. Conclusions

Alcohol acted as a significant treatment-effect modifier for DFS in HPV-positive OPSCC patients, diluting the favorable prognostic effect of HPV positivity. A heavy smoking age of ≥30 pack-years was an independent adverse prognostic factor for OS and DFS in OPSCC patients. The 3-year absolute risk of disease relapse reached up to 50% in HPV-positive OPSCC patients when alcohol use and a heavy smoking age of ≥30 pack-years were combined. The presence of alcohol use and a history of heavy smoking should be considered critical factors when making treatment decisions between standard and de-intensification protocols among HPV-positive OPSCC patients. Further large-scale studies are warranted to confirm these findings.

## Figures and Tables

**Figure 1 jcm-11-06510-f001:**
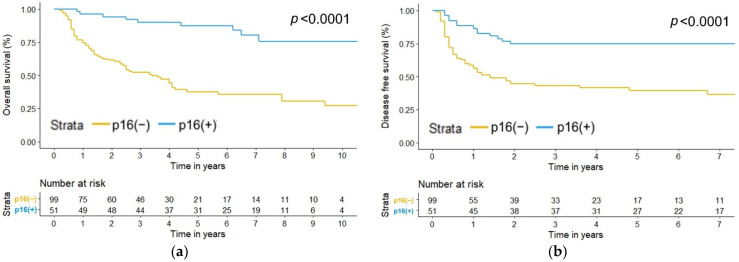
Kaplan–Meier estimate of (**a**) overall survival (*p* < 0.0001) and (**b**) disease-free survival (*p* < 0.0001) by p16 status.

**Figure 2 jcm-11-06510-f002:**
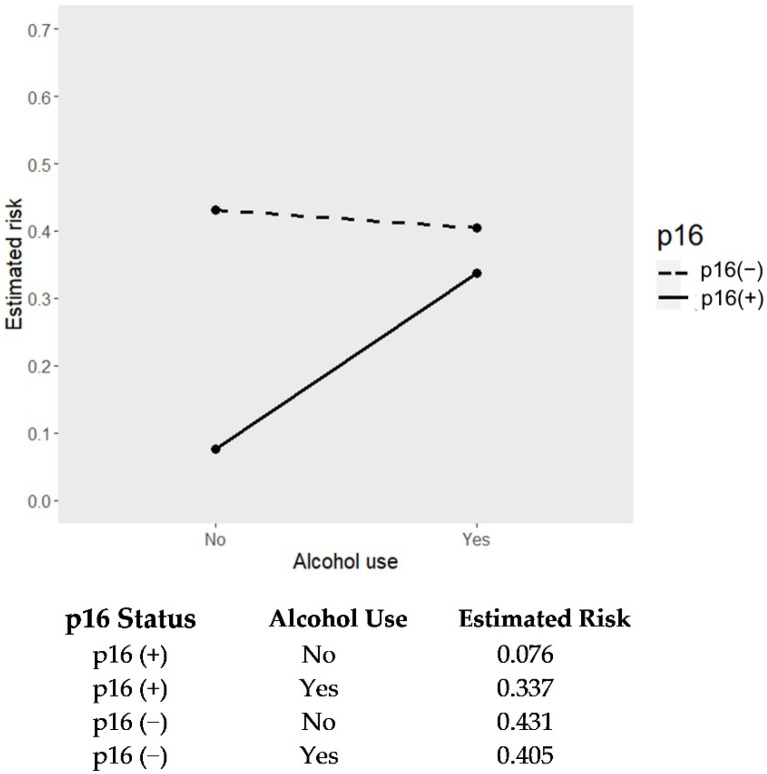
Interaction plot for estimated risk of disease relapse according to p16 status and alcohol use.

**Table 1 jcm-11-06510-t001:** Baseline characteristics of the study population.

Variable	p16 (−), *n* = 99		p16 (+), *n* = 51		*p*-Value
	*n*	%	*n*	%	
Male, *n*, %	93	93.94	37	72.55	<0.001
Female, *n*, %	6	6.06	14	27.45	<0.001
Age, mean, SD	53.3	9.22	56.7	9.71	0.036
Age, *n*, %					0.085
<50	36	36.36	10	19.61	
50–59	35	35.35	20	39.22	
≥60	28	28.28	21	41.18	
Cigarette smoking, *n*, %	91	91.92	26	50.98	<0.001
Smoking age (pack-years), mean, SD	31.8	26.8	14.4	17.9	<0.001
Smoking age (pack-years), *n*, %					<0.001
0	8	8.08	25	49.02	
1–9	7	7.07	3	5.88	
10–19	14	14.14	0	0.00	
20–29	18	18.18	10	19.61	
≥30	52	52.53	13	25.49	
Alcohol use, *n*, %	86	86.87	26	50.98	<0.001
Betel quid chewing, *n*, %	76	76.77	10	19.61	<0.001
ABC concomitant use, *n*, %					<0.001
3	70	70.7	9	17.65	
2 of 3	20	20.2	12	23.53	
Tumor subsite, *n*, %					0.002
Tonsil	47	47.47	40	78.43	
Soft palate	22	22.22	6	11.76	
Tongue base	23	23.23	5	9.80	
Posterior pharyngeal wall	7	7.07	0	0.00	
Clinical stage (AJCC 7th ed.), *n*, %					0.041
Stage I	2	2.02	1	1.96	
Stage II	7	7.07	0	0.00	
Stage III	6	6.06	9	17.65	
Stage IVA	68	68.69	37	72.55	
Stage IVB	16	16.16	4	7.84	
Clinical stage (AJCC 8th ed.), *n*, %					<0.001
Stage I	2	2.02	26	50.98	
Stage II	7	7.07	15	29.41	
Stage III	5	5.05	10	19.61	
Stage IVA	55	55.56	0 *	0.00 *	
Stage IVB	30	30.3	0 *	0.00 *	
Initial treatment, *n*, %					0.024
Surgery	32	32.32	7	13.73	
RT-based therapy	67	67.68	44	86.27	
CCRT	66	66.67	40	78.43	
RT only	1	1.01	4	7.84	
Disease relapse, *n*, %	58	58.59	13	25.49	<0.001
LRR	25	25.25	6	11.76	
DM	18	18.18	5	9.80	
LRR + DM	15	15.15	2	3.92	
Mortality, *n*, %	61	61.62	9	17.65	<0.001
DOD	47	47.47	5	9.80	
Dead, other reason	14	14.14	4	7.84	

* The clinical stage of nonmetastatic HPV-positive OPSCC was downstaged to stage III or less in the eighth edition of the American Joint Committee on Cancer (AJCC) staging system. Abbreviations: *n*, number of patients; SD, standard deviation; ABC, alcohol/betel nut/cigarette; AJCC, American Joint Committee on Cancer; ed., edition; RT, radiotherapy; CCRT, concurrent chemoradiotherapy; LRR, locoregional recurrence; DM, distant metastasis; DOD, died of disease.

**Table 2 jcm-11-06510-t002:** Univariate and multivariate analyses for overall survival.

Variable	Univariate		Multivariate			
			Stepwise Selection *		Selected Predictors ^#^	
	HR	95% CI	HR	95% CI	HR	95% CI
**Covariate**						
p16 (ref. = negative)	0.18	(0.09, 0.37)	0.08	(0.02, 0.35)	**0.09**	**(0.02, 0.44)**
Gender (ref. = male)	0.15	(0.04, 0.60)			0.52	(0.12, 2.33)
Age (5-year increments)	0.89	(0.78, 1.02)				
Clinical stage (ref. = stage I–III) ^&^						
Stage IVA	2.69	(1.07, 6.76)	2.55	(1.00, 6.49)	**2.72**	**(1.05, 7.00)**
Stage IVB	12.11	(4.41, 33.26)	16.26	(5.76, 45.86)	**15.62**	**(5.29, 46.13)**
Tumor subsite (ref. = other sites than tonsil)	0.46	(0.28, 0.73)			0.77	(0.45, 1.33)
Initial treatment (ref. = surgery)	0.88	(0.52, 1.48)			1.02	(0.58, 1.81)
Smoking age (ref. = <20 pack-years)						
20–29	1.78	(0.85, 3.70)	0.96	(0.41, 2.23)	1.15	(0.46, 2.90)
≥30	2.98	(1.67, 5.32)	1.90	(1.02, 3.54)	**2.05**	**(1.05, 4.00)**
Alcohol use (ref. = none)	3.19	(1.58, 6.44)			0.90	(0.39, 2.08)
Betel quid chewing (ref. = none)	2.35	(1.40, 3.96)			0.85	(0.45, 1.61)
**Interaction term**						
p16: Smoking age (20–29 pack-years)			8.00	(1.16, 55.01)	5.74	(0.79, 41.53)
p16: Smoking age (≥30 pack-years)			2.94	(0.43, 19.93)	2.53	(0.36, 17.85)
p16: Alcohol use						
p16: Betel quid chewing						

* Variable selection employed the stepwise method by the Akaike information criterion. ^#^ The Cox proportional hazard model was constructed with interaction terms that were selected by the stepwise method and significant and clinically relevant variables from univariate analyses. ^&^ The clinical stage was defined by the 7th edition of the American Joint Committee on Cancer (AJCC) staging system. Significant values of HR and 95% CI are in bold. Abbreviations: HR, hazard ratio; CI, confidence interval; ref., reference.

**Table 3 jcm-11-06510-t003:** Univariate and multivariate analyses for disease-free survival.

Variable	Univariate		Multivariate			
			Stepwise Selection *		Selected Predictors ^#^	
	HR	95% CI	HR	95% CI	HR	95% CI
**Covariate**						
p16 (ref. = negative)	0.31	(0.17, 0.56)	0.12	(0.03–0.58)	**0.10**	**(0.02, 0.49)**
Gender (ref. = male)	0.32	(0.12, 0.88)			1.12	(0.36, 3.42)
Age (5-year increments)	0.98	(0.87, 1.11)	1.12	(0.97–1.28)		
Clinical stage (ref. = stage I–III) ^&^						
Stage IVA	2.75	(1.09, 6.90)	3.6	(1.37–9.45)	**2.87**	**(1.11, 7.41)**
Stage IVB	10.68	(3.88, 29.41)	12.35	(4.09–37.26)	**8.43**	**(2.83, 25.08)**
Tumor subsite (ref. = other sites than tonsil)	0.34	(0.21, 0.55)	0.49	(0.26–0.83)	**0.46**	**(0.27, 0.80)**
Initial treatment (ref. = surgery)	1.05	(0.62, 1.78)			1.21	(0.68, 2.16)
Smoking age (ref. = <20 pack-years)						
20–29	1.43	(0.71, 2.88)	1.69	(0.77–3.70)	1.64	(0.74, 3.63)
≥30	2.07	(1.20, 3.55)	1.88	(1.01–3.49)	**1.99**	**(1.06, 3.75)**
Alcohol use (ref. = none)	2.68	(1.37, 5.23)	0.66	(0.27–1.62)	0.57	(0.23, 1.39)
Betel quid chewing (ref. = none)	1.77	(1.08, 2.92)			0.91	(0.48, 1.72)
**Interaction term**						
p16: Smoking age (20–29 pack-years)						
p16: Smoking age (≥30 pack-years)						
p16: Alcohol use			4.4	(0.78–24.7)	**6.00**	**(1.03, 35.01)**
p16: Betel quid chewing						

* Variable selection employed the stepwise method by the Akaike information criterion. ^#^ The Cox proportional hazard model was constructed with interaction terms that were selected by the stepwise method and significant and clinically relevant variables from univariate analyses. ^&^ The clinical stage was defined by the 7th edition of the American Joint Committee on Cancer (AJCC) staging system. Significant values of HR and 95% CI are in bold. Abbreviations: HR, hazard ratio; CI, confidence interval; ref., reference.

**Table 4 jcm-11-06510-t004:** The cumulative risks of disease relapse in HPV-related oropharyngeal cancer by alcohol and smoking age.

Smoking Age	Follow-Up	p16 (+)	
		Absolute Risk (95% CI)	Absolute Risk (95% CI)
		Alcohol (−)	Alcohol (+)
<20 pack-years	3 years	0.12 (0.11, 0.14)	0.33 (0.24, 0.39)
	5 years	0.14 (0.12, 0.15)	0.35 (0.26, 0.42)
20–29 pack-years	3 years	0.19 (0.18, 0.20)	0.45 (0.35, 0.51)
	5 years	0.21 (0.20, 0.21)	0.47 (0.37, 0.54)
≥30 pack-years	3 years	0.22 (0.21, 0.24)	0.50 (0.38, 0.57)
	5 years	0.24 (0.22, 0.25)	0.52 (0.41, 0.59)

Abbreviations: HPV, human papilloma virus; CI, confidence interval.

## Data Availability

The data presented in this study are available on reasonable request from the corresponding author. The data are not publicly available due to privacy.

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
