# Peer review of "Impact of Alcohol and Smoking on Outcomes of HPV-Related Oropharyngeal Cancer"

_jcm, 2022, doi:10.3390/jcm11216510_

Round 1

Reviewer 1 Report

Authors have provided important association of Alcohol and Smoking on Outcomes of HPV-related  Oropharyngeal  Cancer. Manuscript requires consideration to address following comments

1) Table 1, total of n = 150 patients were mentioned and only demographic information for male subjects (n=130) were provided. Please provide information for total n = 150 patients. Is there any specific reason to exclude the female participants in the present study?

2) Discuss and compare the findings of present study with the study of Anantharaman et al. (International Journal of Epidemiology, 2016) and include same in ‘Discussion’ section of the study.  

3) Discuss the present findings in context with similar studies performed in other populations for e.g. Auguste et al., 2020 (Cancer Med. 2020 Sep; 9(18): 6854–6863) and Kumat et al. 2015 (PLoS One. 2015; 10(10): e0140700).

4) What measures Authors adopted to avoid selection biasness in retrospective study?

Author Response

Response to Reviewer 1 Comments

Many thanks to this reviewer for his/her very valuable comments/suggestions. We have addressed these concerns as described below.

Point 1: Table 1, total of n = 150 patients were mentioned and only demographic information for male subjects (n = 130) were provided. Please provide information for total n = 150 patients. Is there any specific reason to exclude the female participants in the present study?

Response 1: Table 1 demonstrated all the 150 patients’ characteristics. The variable “Male, n, %” in table 1 indicated the number and proportion of male subjects in total of 150 patients, and we did not exclude the female participants. Besides, all 150 patients including male and female participants were analyzed in the present study. We thank the reviewer’s comments and have revised table 1.

Added detailed information of female subjects (n = 20) in Table 1.

Point 2: Discuss and compare the findings of present study with the study of Anantharaman et al. (International Journal of Epidemiology, 2016) and include same in ‘Discussion’ section of the study.

Response 2: Anantharaman et al. conducted the largest study by pooling two large head and neck cancer studies with HPV serology data to examine the relationship between smoking and HPV infection for oropharyngeal cancer (OPC). The result demonstrated that smoking was consistently associated with an increase in OPC risk in models stratified by HPV16 seropositivity. In addition, the prevalence of OPC increased with smoking for both HPV16-positive and HPV16-negative patients. We agreed that tobacco exposure was an important risk factor for OPC irrespective of HPV status, and this conclusion was supported by previous studies. However, we further evaluate the impact of these risk factors on the treatment outcomes of OPC. Our findings showed that heavy smoking was an independent adverse prognostic factor of overall survival (OS) and disease-free survival (DFS) in OPC patients, regardless of HPV status. These results implied that the impact of smoking on OPC may have important implications for carcinogenesis, treatment, survival, and disease relapse.

Added: “By pooling two large head and neck cancer studies with HPV serology data, Anantharaman et al. demonstrated that smoking was consistently associated with increased risks of both HPV-positive and HPV-negative OPSCC [31].” In the Discussion section, paragraph 4, the 7th to 10th sentences.

Point 3: Discuss the present findings in context with similar studies performed in other populations for e.g. Auguste et al., 2020 (Cancer Med. 2020 Sep; 9(18): 6854–6863) and Kumat et al. 2015 (PLoS One. 2015; 10(10): e0140700).

Response 3: These two studies investigated the role of tobacco and alcohol consumption on the occurrence of head and neck cancer, and the association of these factors with oral HPV infection in non-Western populations. Both studies agreed that tobacco and alcohol act synergistically in the process of HNC development and progression, but the synergistic effect was substantially lower in HPV-positive than in HPV-negative HNSCC in Afro-Caribbean population. We thank the reviewer’s comments and have revised the introduction.

Added references: “Although the role of ABC lifestyle factors has been well established in the development of HNSCC [18,19], less is known about their prognostic significance in patients with HPV-positive OPSCC.” In the Introduction section, paragraph 3, sentence 10.

Point 4: What measures Authors adopted to avoid selection biasness in retrospective study?

Response 4: From January 2010 to October 2019, we consecutively collected 150 OPC patients who had received a complete staging work-up, a complete course of curative treatment, and adequate follow-up in our hospital. The OPC patient number diagnosed in our hospital (a medical center in Southern Taiwan) was around 20-30 persons per year. The exclusion criteria were 1) other underlying malignancy or distant metastasis at the time that OPC was diagnosed; 2) lack of available pretreatment primary tumor specimens to re-evaluate the p16 expression by IHC staining; 3) lack of pretreatment contrast-enhanced computed tomography (CT) images of the head and neck regions to re-evaluate the clinical staging. Therefore, the internal validity of patient selection was solid and reliable. However, the main limitation of the present study was its retrospective design and relatively small sample size in a single institution; a larger sample size may have had sufficient power to convince the conclusion. We appreciate this reviewer for his (her) very constructive comments. The revised version has rephrased that sentence.

Added: “Despite the retrospective study design, the consecutive enrollment of qualified OPSCC participants made the internal validity of patient selection solid and reliable.” In the Discussion section, paragraph 5, the 10th to 12th sentences.

Please see the attachment for the revised manuscript.

Reviewer 2 Report

The authors present a well conducted analysis of effect modifiers for OPSCC in a southeast asian population. The overall conclusions from this paper are not dramatically novel, however lend insight into individual factors which may impact outcome in HPV-associated OPSCC in a less studied demographic. The study is well conducted statistically and I have only minor comments.

While a busy table, inclusion of p-values in tables 2-4 would be recommended. 

Author Response

Response to Reviewer 2 Comments

Many thanks to this reviewer for his/her very valuable comments/suggestions. We have addressed these concerns as described below.

Point 1: The authors present a well conducted analysis of effect modifiers for OPSCC in a southeast Asian population. The overall conclusions from this paper are not dramatically novel, however lend insight into individual factors which may impact outcome in HPV-associated OPSCC in a less studied demographic. The study is well conducted statistically and I have only minor comments.

While a busy table, inclusion of p-values in tables 2-4 would be recommended.

Response 1: We appreciate this reviewer for his (her) very friendly comments. We consider 95% confidence interval is more reasonable for hazard ratios and estimated risk in this study. According to Oliver Frank (2021) published paper, statistical significance only means that the data reached an arbitrarily defined level of incompatibility with the statistical model, but it cannot reveal how precise the estimation is. In our study, we did not only investigate the association between significant predictors and overall survival/disease-free survival, but also estimate the risk with different follow-up times in these predictors. Therefore, the precision of risk estimation is more important than statistical significance. The 95% confidence interval is an indicator to present the precision of estimation (the smaller 95% confidence interval is more precise). In addition, 95% confidence interval also can tell readers whether the predictors are significant (the lowest 95% confidence interval is above 1). Thus, we consider 95% confidence intervals would be a better choice for our study as compared to p-values.

Reference: Frank O, Tam CM, Rhee J. Is it time to stop using statistical significance? Australian prescriber. 2021;44(1):16-18.
